# Description of a New Species of *Parathalestris* Brady & Robertson, 1873 (Copepoda: Harpacticoida: Thalestridae) from China, with a Key to Its Affiliated Species Group †

**Yihong Wu** [1,2,‡,§], **Lin Ma** [1,3,‡,§] **and Qi Kou** [1,2,3,*,§]

1   Institute of Oceanology, Chinese Academy of Sciences, Qingdao 266071, China
2   University of Chinese Academy of Sciences, Beijing 100039, China
3   Laboratory for Marine Biology and Biotechnology, Pilot National Laboratory for Marine Science and Technology, Qingdao 266237, China
*   Correspondence: kouqi@qdio.ac.cn
†   urn:lsid:zoobank.org:pub:037A53E3-8E60-474B-BD67-08B2E25641FC
‡   These authors contributed equally to this work.
§   urn:lsid:zoobank.org:author:8C25BFF8-F75B-4778-AFD9-9A3465B760E8 (Y.W.);
    urn:lsid:zoobank.org:author:1979265D-665C-4572-AB98-EA5731D5D87C (L.M.);
    urn:lsid:zoobank.org:author:4FCB5C5A-5F44-458D-B8CE-B861D51ACDB2 (Q.K.)

**Abstract:** A new species belonging to the genus *Parathalestris* Brady & Robertson, 1873 was identified and described here based on samples recently collected from a survey on fouling organisms at the Zhonggang wharf in Qingdao, China. The new species differs from its congeners in the following characteristics: caudal rami 2.0 times longer than wide, with bulbous setae II and setae III of caudal ramus issuing terminally and laterodistally, respectively; exopod of antenna two-segmented, with two setae on first segment and four setae on second one;maxilla with two bipinnate setae on proximal endite, and three spinulose setae on middle and distal endites; basis of maxilliped with concave inner edge; P1 exopod and endopod approximately same length, enp-1 about 6.7 times as long as greatest width, with inner seta reaching proximal 2/3 of segment; baseoendopod and exopod of female P5 with transversal rows of spinules on its surface, exopod bearing six marginal setae; male P1 carrying an inner curved spine on basis; exopod of male P5 bearing six elements. The DNA barcode (COI) sequence of the new species was obtained and submitted to GenBank. This is the first report of *Parathalestris* from the China Seas.

**Keywords:** Crustacea; littoral; meiofauna; Qingdao; taxonomy; Thalestrinae





## 1. Introduction

The genus *Parathalestris* Brady & Robertson, 1873 belongs to the harpacticoid copepod family Thalestridae Sars, 1905, and comprises 28 valid species [1–3]. As the most speciose genus of the Thalestridae, *Parathalestris* is characterized by the morphological combination of the two-segmented exopod of the antenna, the second segment of the P2 endopod with two inner setae, and both the enp-2 of P3 and P4 with only one inner seta [4].

Species of the genus *Parathalestris* are benthic forms that mainly inhabit the surface of sediment [5]. Most of the species are associated with macrophytes, but only *P. infesta* Ho & Hong, 1988 has been formally reported from mine algal fronds [6,7]. At present, eight species from the West Pacific have been reported, mainly concentrated in eastern Russia, Japan, South Korea and Thailand [1–4,6,8–11].

The diversity and taxonomy of the marine harpacticoid family Thalestridae are still rarely studied in China. So far, only three species of two other genera of the family (*Eudactylopus andrewi* Sewell, 1940, *Phyllothalestris mysis* (Claus 1863), *P. sarsi* Sewell, 1940) have been recorded from the China Seas in ecological research data, yet without detailed descriptions [12,13]. Recently, biological samples adhering to the testing panels were

collected during a survey of fouling organisms at the Zhonggang wharf in Qingdao. Marine fouling organisms refers to the microorganisms, plants and animals sessile to or perching on marine facilities, which are harmful to aquaculture [14]. In the laboratory, a number of harpacticoid copepods were isolated from these samples. After careful examination, the presence of a new species of *Parathalestris* was confirmed, which is described and illustrated here. Further morphological comparison suggested the new species can be assigned to the species group I according to Huys [7]. The new species represents not only the first occurrence of *Parathalestris* from fouling organisms, but also the first report of the genus from China.

## 2. Material and Methods

Samples were collected from testing panels at the Zhonggang wharf in Qingdao. Specimens were fixed in 100% alcohol. The harpacticoid copepod specimens were isolated from the samples under a light microscope (Nikon SMZ1270). Before DNA extraction, the specimens were preserved in 100% alcohol and refrigerated at −20 °C. The genomic DNA was extracted prior to further morphological examination.

Each specimen was transferred to a 1.5 mL sterilized centrifuge tube containing 180 μL of ATL buffer and 20 μL of Proteinase K for non-destructive DNA extraction [15]. The subsequent DNA extraction process was performed according to the protocol of QIAamp DNA Micro Kit (Qiagen, Hilden, Germany), except that the exoskeleton of each specimen was picked out and transferred into a new 1.5 mL sterilized centrifuge tube containing sterile distilled $H_2O$ after the specimen was completely digested. Polymerase chain reaction (PCR) amplification was carried out in a reaction mix with 3 μL of the DNA, 12.5 μL of Premix Taq$^{TM}$ (Takara, Otsu, Shiga, Japan), 1 μL of each primer (10 mM) and 7.5 μL sterile distilled $H_2O$. The mitochondrial COI gene was amplified using the primer LCO1490/HCO2198 [16]. The PCR conditions were as follows: initial denaturation for 5 min at 94 °C, followed by 45 cycles of denaturation at 94 °C for 60 s, annealing at 48 °C for 90 s, extension at 72 °C for 60 s, and a final extension at 72 °C for 5 min. PCR products were purified using the Wizard$^{TM}$ SV Gel and PCR Clean-UP System (Promega, Madison, WI, United States) before sequencing. The purified PCR products were sequenced from both directions using the same forward and reverse primers for PCR amplification with ABI 3730XL DNA Analyzer (Applied Biosystems, Foster City, CA, United States). Each sequence was BLASTed in the NCBI database to confirm it was not contaminated. Then, the DNA barcode sequence was submitted to GenBank.

After DNA extraction, the exoskeleton of each specimen and other specimens preserved in 100% alcohol was dissected and observed under a stereomicroscope (Nikon SMZ1270). Before dissection, the habitus was drawn, and the body length was measured. Specimens were dissected in lactic acid and mounted on slides, which were subsequently coated by coverslips and sealed with nail polish. Observations and drawings of body parts were made using a differential interference contrast microscope (Nikon Eclipse Ni) equipped with a drawing tube. Most illustrations were drawn at 400× and 600× magnification, except the mouthparts which were drawn at 1000× magnification, with an oil immersion lens.

The descriptive terminology follows that of Huys et al. [17]. Abbreviations used in the text and figures are as follows: *ae*, aesthetasc; *A2*, antenna; *P1–P6*, first to sixth thoracic legs; *exp (enp)-1(-2, -3)*, the proximal (middle, distal) segment of an exo- or endopod. The body length was measured from the anterior margin of the rostrum to the posterior margin of the caudal rami. The type material is deposited in the Marine Biological Museum, Chinese Academy of Sciences, Qingdao, China (MBMCAS).

We sampled 12 morphological characteristics for the phylogenetic analysis of the species group I of *Parathalestris*, with *P. yeemini* Chullasorn, Kangtia and Song, 2016 (species group III) used as an outgroup. Because the description of *P. similis* Lang, 1936 and *P. vinosa* Pallares, 1975 are not detailed enough to be discriminated from *P. irelandica* Roe, 1958, these

three species were not included in the analysis. The following 12 morphological characters and corresponding states (in parentheses) are listed:

1.  Length-width ratio of female caudal ramus: (0) $\leq$ 1; (1) > 1.
2.  Bulbous seta on caudal ramus: (0) absent; (1) present.
3.  Armature formula of A2 exopod: (0) (1:1, 3); (1) (2:1, 2); (2) (2:1, 3).
4.  Inner edge of maxilliped basis: (0) straight; (1) concave.
5.  Length-width ratio of P1 endopod: (0) $\leq$ 5; (1) > 5.
6.  Rows of transversal spinules on female P5 baseoendopod: (0) absent; (1) present.
7.  Length of female P5: (0) extends to half-length but not reaching distal margin of the genital double-somite; (1) reaching distal margin of genital double-somite; (2) exceeding distal margin of genital double-somite, close to penultimate somite.
8.  Length of female P5 baseoendopod: (0) not exceeding distal margin of exopod; (1) exceeding distal margin of exopod.
9.  Shape of inner spine on male P1 basis: (0) normal; (1) curved.
10. Armature formula of male P5: (0) 3/5; (1) 3/6; (2) 2/5.
11. Median seta on male P5: (0) present, weak; (1) present, normal; (2) absent.
12. Length of outermost seta on male P5 baseoendopod: (0) equal to median one; (1) half-length of median one; (2) less than half-length of median one.

A character matrix was constructed (Table 1) to contain all character states for the 10 ingroup and 1 outgroup species. Unavailable or unknown character states were marked as "–". The analysis was implemented in IQ-Tree2 v.2.1.2 [18] using morphological ML (MK) and ascertainment bias correction (ASC) models [19] with all characters unweighted.

**Table 1.** Characteristic states scored for 10 species of the species group I of *Parathalestris* and 1 outgroup, *Parathalestris yeemini* Chullasorn, Kangtia & Song, 2016. Unavailable or unknown characters states are marked with a dash ("–").

| Characters | 1 | 2 | 3 | 4 | 5 | 6 | 7 | 8 | 9 | 10 | 11 | 12 |
|---|---|---|---|---|---|---|---|---|---|---|---|---|
| *P. areolata* | 1 | 1 | 2 | 1 | 0 | 1 | 0 | 0 | 1 | 1 | 1 | 2 |
| *P. bulbiseta* | 0 | 1 | 1 | 1 | 0 | 0 | 0 | 0 | 1 | 1 | 1 | 2 |
| *P. cambriensis* | 0 | 0 | 1 | 0 | 1 | 0 | 0 | 1 | 1 | 1 | 1 | 1 |
| *P. croni* | 1 | 0 | 0 | 0 | 0 | 0 | 0 | 0 | 1 | 1 | – | – |
| *P. dovi* | 0 | 0 | 2 | 0 | 1 | 0 | 0 | 0 | 1 | 1 | 1 | 2 |
| *P. harpactoides* | 0 | 0 | 1 | 0 | 1 | 0 | 0 | 0 | – | 1 | 1 | 1 |
| *P. jacksoni* | 1 | 0 | 1 | 1 | 0 | 0 | 0 | 1 | – | – | – | – |
| *P. jejuensis* | 0 | 0 | 2 | 0 | 0 | 0 | 1 | 0 | 0 | 0 | 0 | 1 |
| *P. parviseta* | 0 | 0 | 2 | 0 | 0 | 1 | 1 | 0 | 1 | 0 | 0 | 0 |
| *P. xinzhengi* **sp. nov.** | 1 | 1 | 2 | 1 | 1 | 1 | 1 | 1 | 1 | 1 | 1 | 2 |
| *P. yeemini* | 0 | 0 | 2 | 0 | 0 | 1 | 2 | 1 | 0 | 2 | 2 | – |

## 3. Results

### 3.1. Systematics

Order: Harpacticoida Sars G.O., 1903
Family: Thalestridae Sars G.O., 1905
Subfamily: Thalestrinae Sars, 1905
Genus: *Parathalestris* Brady & Robertson, 1873
Type species. *Parathalestris clausi* (Norman, 1869)
*Parathalestris xinzhengi* **sp. nov.**
(Figures 1–5)



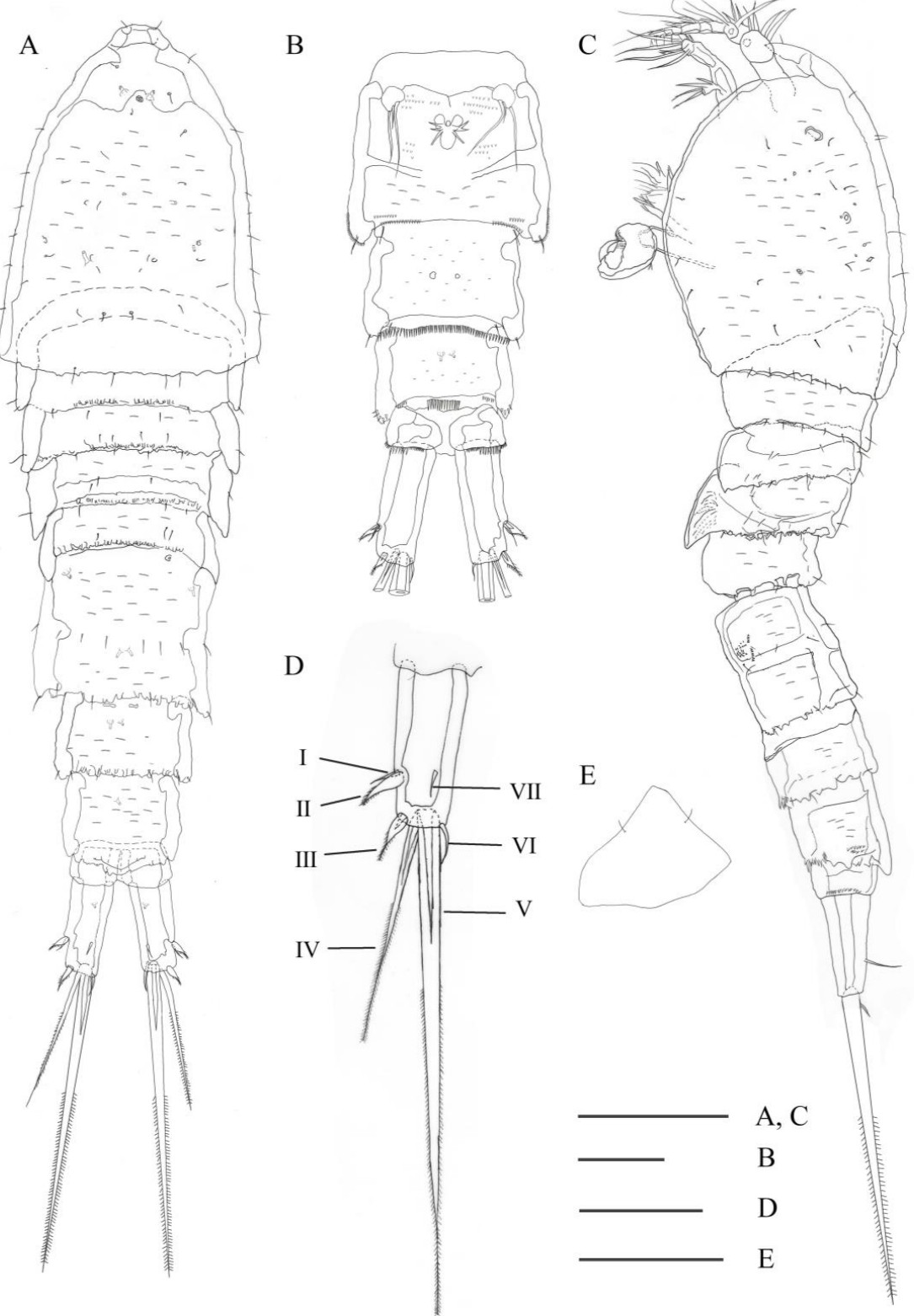

**Figure 1.** *Parathalestris xinzhengi* **sp. nov.**, female: (**A**) Habitus, dorsal. (**B**) Urosome, ventral. (**C**) Habitus, lateral. (**D**) Caudal ramus, with the setae labelled with I–VII. (**E**) Rostrum, dorsal. Scale bars: 20 μm (**A**,**C**); 100 μm (**B**); 10 μm (**D**); 100 μm (**E**).

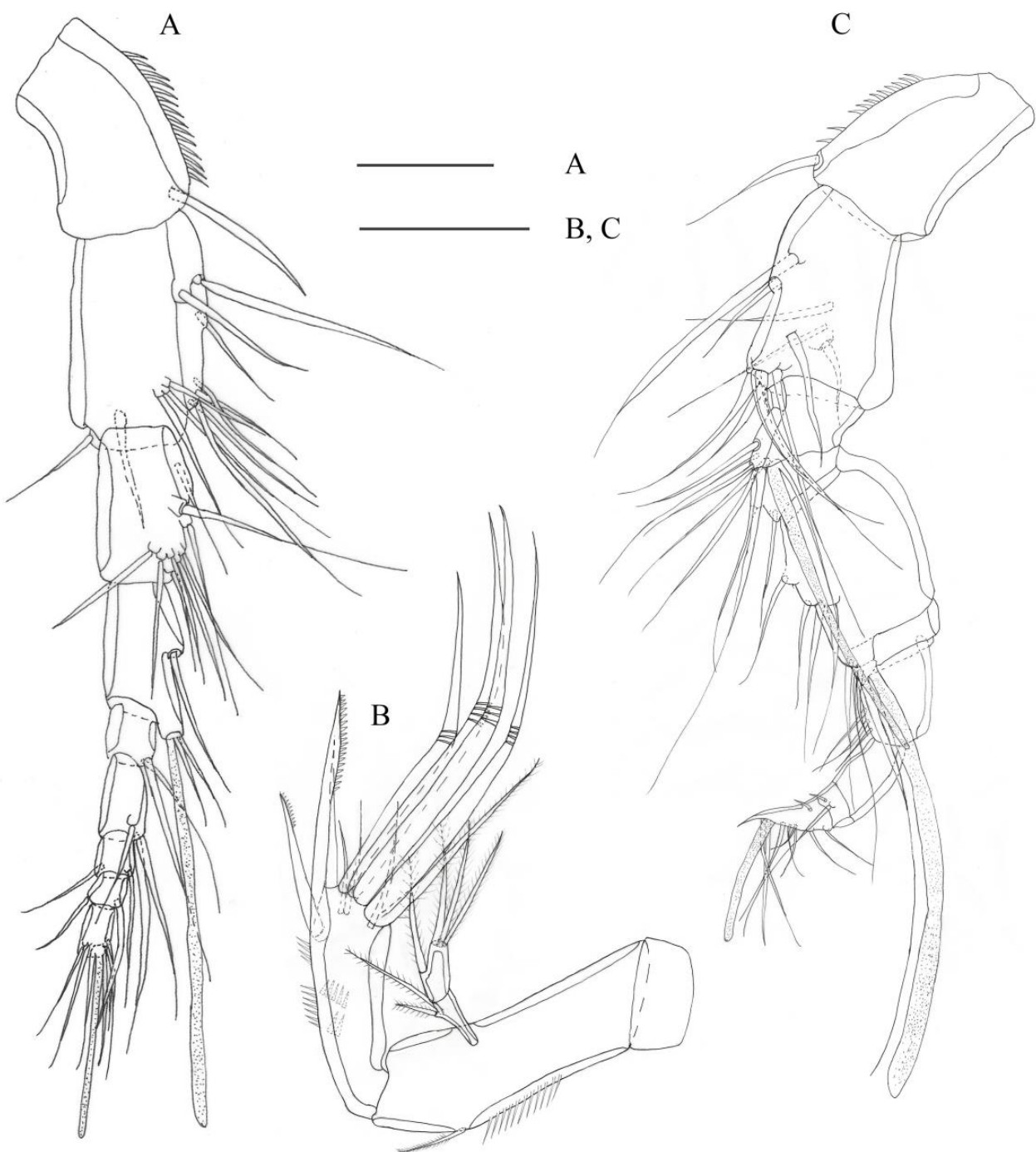

**Figure 2.** *Parathalestris xinzhengi* **sp. nov.** (**A**) Antennule of female. (**B**) Antenna. (**C**) Antennule of male. Scale bars: 20 μm (**A**); 50 μm (**B**,**C**).

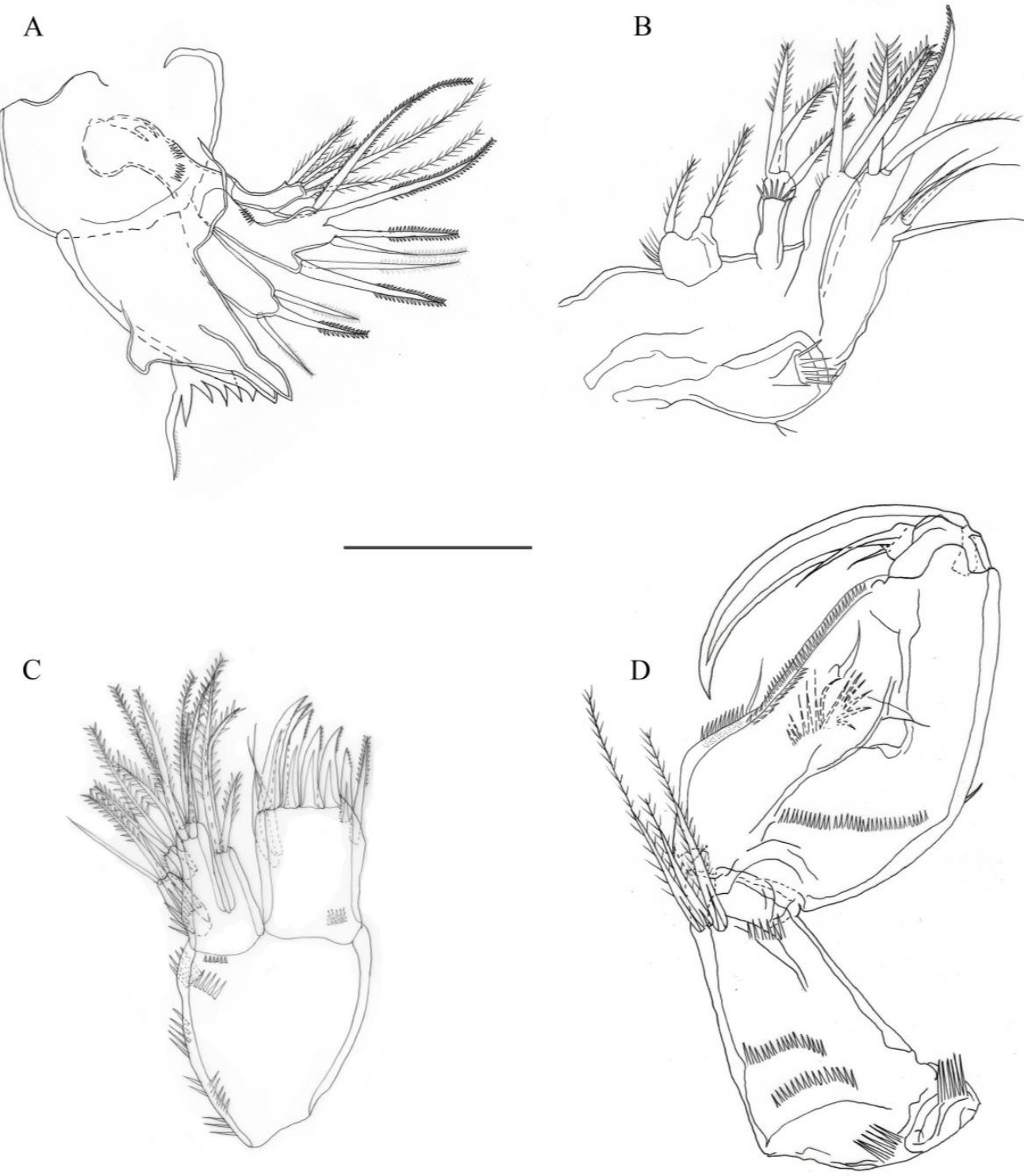

**Figure 3.** *Parathalestris xinzhengi* **sp. nov.** (**A**) Mandible. (**B**) Maxilla. (**C**) Maxillule. (**D**) Maxilliped. Scale bar: 50 μm.

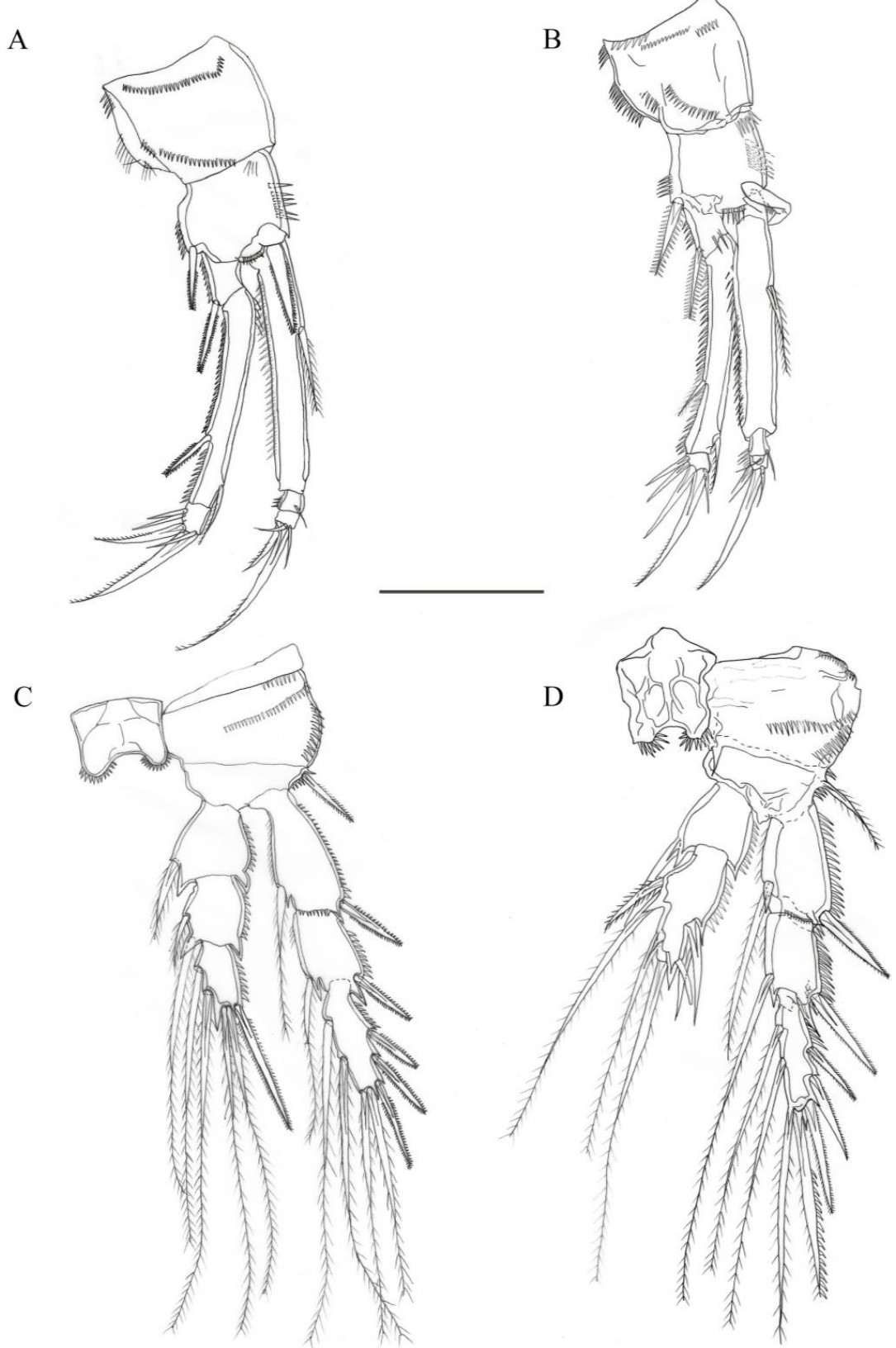

**Figure 4.** *Parathalestris xinzhengi* **sp. nov.** (**A**) P1 female. (**B**) P1 male. (**C**) P2 female. (**D**) P2 male. Scale bar: 100 μm.

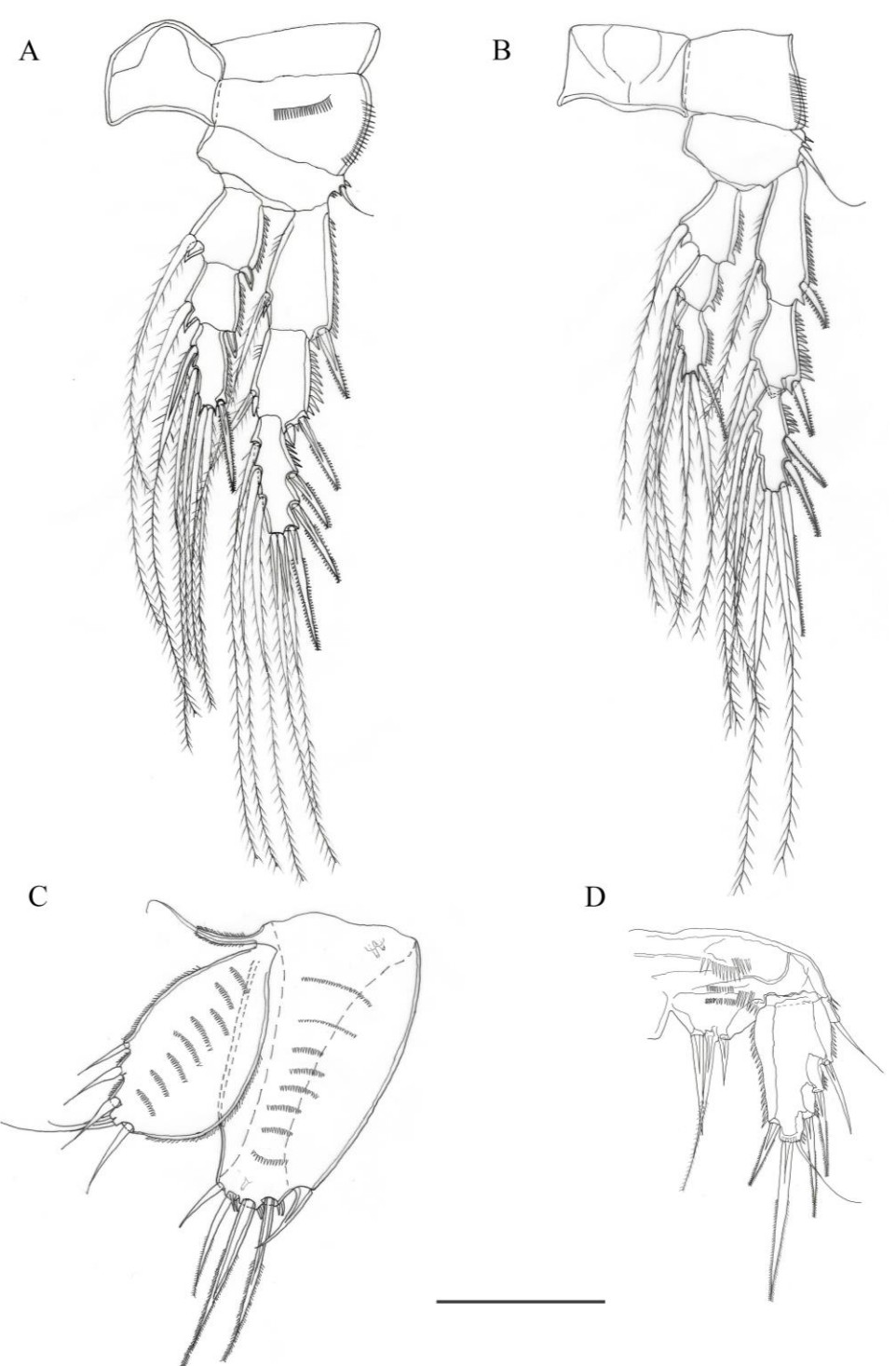

**Figure 5.** *Parathalestris xinzhengi* **sp. nov.** (**A**) P3 female. (**B**) P4 female. (**C**) P5 female. (**D**) P5 male. Scale bar: 100 μm.

urn:lsid:zoobank.org:act:67E3CEF2-01E2-4B95-AF64-4E188DC884E8

**Type locality.** Zhonggang wharf, Qingdao (36°1′ N, 120°1′ E), 0.8 m depth, collected by Dr. Huilian Liu, May 2018.

**Material examined.** Holotype: 1♀, dissected on 3 slides (MBM 189268). Allotype: 1♂, for DNA sequencing and dissected on 7 slides (MBM 189269, amplification successful, GenBank accession number: OQ831995). Paratypes: 1♀, dissected on 4 slides (MBM 189270); 1♀, partly dissected on 2 slides (MBM 189271); 1♂, dissected on 3 slides (MBM 189272); 1♂ for DNA sequencing and preserved in 70% alcohol with some drops of glycerin (MBM

189273, amplification successful); 3♀♀, not dissected, preserved in 70% alcohol with some drops of glycerin (MBM 189274). All specimens were collected from the type locality.

**Etymology.** The species is named after Dr. Xinzheng Li, principal investigator at IOCAS, for his great contributions to the marine invertebrate diversity research in China, with QK's gratitude for his mentoring and guidance.

### 3.2. *Description*

3.2.1. Description of Female

Total body length ranging from 846 to 1348 μm (mean = 1002 μm; *n* = 4), greatest width ranging from 235 to 344 μm (mean = 268 μm; *n* = 4).

Habitus (Figure 1). Body long and fusiform, widest at the posterior margin of cephalothorax, tapering posteriorly. All somites with sensilla as illustrated, except for the anal somite. Posterior edges of all somites except for cephalothorax and anal somite are furnished with some projections dorsally. Prosome four-segmented; cephalothorax (including two thoracic somites bearing maxilliped and P1) almost as long as three articulated somites combined, the latter bearing P2 to P4, ornamented with rows of microspinules scattered on dorsal surface. Urosome five-segmented, comprising P5-bearing somite, genital double-somite and three free abdominal somites. Third and fourth urosomites with pores on dorsal and ventral surfaces, ventrally with a row of spinules on posterior edge. Genital double-somite fused midventrally, with pores on dorsal surface, denticles on posterolateral edge and some fine spinules on ventral surface; genital field distinct, ventrally with two copulatory apertures and P6, the latter represented by three smooth setae of unequal length (median one minute). Anal operculum smooth, with some projections on posterior edge. Caudal ramus long, twice as long as wide, with seven elements: seta I and II issuing at distal third of ramus, the former shorter than the latter, seta I naked and seta II bulbous; setae III, IV, V and VI located apically; seta III bulbous; setae IV and V bipinnate, well-developed and fused basally, seta V about twice as long as seta IV, while other five setae small and tiny; seta VI naked and slender; seta VII naked, located on inner proximal portion of dorsal surface.

Rostrum (Figure 1E). Small, articulated with cephalothorax, directed downward, almost triangular with pair of sensilla mid-dorsally.

Antennule (Figure 2A). With nine segments, proximal four segments long; first segment ornamented with row of spinules; surface of other eight segments smooth. Aesthetasc on fourth segment reaching far beyond distal end of the last segment. Armature formula (all antennular setae here are bare): 1-[1], 2-[12], 3-[9], 4-[(2 + (1+ae))], 5-[2], 6-[3], 7-[2], 8-[2], 9-[4 + (2+ae)].

Antenna (Figure 2B). Coxa short. Allobasis elongate, ornamented with row of spinules as figured and with one plumose abexopodal seta in distal third of anterior edge. Endopod one-segmented, shorter than allobasis, with rows of spinules on inner margin and subapically, bearing eleven elements: two unipinnate spines laterodistally, two slender setae on subapical surface, one spine, four geniculate, one naked and one plumose setae distally. Exopod two-segmented, first segment with two plumose setae, second segment with one lateral and three apical plumose setae.

Mandible (Figure 3A). Gnathobase well developed, bearing several multicuspidate teeth on three divided distal lobes, and one unipinnate long seta on distal corner. Basis elongate, with three bipinnate setae. Endopod bilobed with two and three pinnate setae, respectively. Exopod shorter than endopod, with one lateral and four distal setae.

Maxillule (Figure 3B). Praecoxa with few spinular rows along outer margin. Arthrite strongly developed, bearing one spinulose row on posterior surface, with six strong apical spines and some spinules on each of the spines of the arthrite, one lateral bipinnate seta and two smooth anterior surface setae. Coxa with cylindrical process bearing three spinose setae. Basis with several spinules on outer margin, bearing five elements. Endopod one-segmented, with three spinose setae. Exopod one-segmented, bearing one naked and two spinose setae.

Maxilla (Figure 3C). Syncoxa with several spinules on outer margin, bearing three endites: proximal endite bilobed with two plumose setae, middle and distal endites with three spinulose setae each. Alloasis drawn out into strong, spinose claw with one plumose and one bipinnate seta. Endopod represented by a small protuberance bearing five setae near base of claw.

Maxilliped (Figure 3D). Syncoxa and basis both developed well. Syncoxa about as long as basis, with several rows of spinules on anterior surface and three plumose setae at inner distal corner. Basis longer than endopod, inner edge convex at proximal third, slightly concave at almost half-length, with one slender seta on inner edge, one seta on surface, and rows of spinules along inner and outer margins. Endopod drawn out into a strong, inwardly curved claw and a conical process bearing two smooth setae.

P1 (Figure 4A). Coxa with two rows of spinules on anterior surface, and few spinules and some setules along outer margin. Basis with row of spinules along inner and outer margins, additionally, with one inner and one outer bipinnate spine. Endopod about as long as exopod; both rami three-segmented. Enp-1 much longer than enp-2 and enp-3 combined, about 6.7 times as long as greatest width, with one plumose seta on proximal third of inner margin; enp-2 as long as enp-3, with several spinules on outer margin, inner margin with one minute seta; enp-3 with one median strong outwardly curved unipectinate spine and one small seta on tip, one outer smooth spine and one inner small smooth seta. All segments of exopod ornamented with fine spinules along outer margin, exp-2 much longer than exp-1 and exp-3; exp-1 slightly longer than exp-3, with one outer bipinnate spine; exp-2 with one outer bipinnate spine on distal third of outer margin and one unipinnate inner seta close to distal margin; exp-3 shortest, with one slender inner seta, two outwardly curved unipectinate spines and two outer naked setae.

P2 (Figure 4C). Intercoxal sclerite with spinules on two lateral blunt projections. Prae-coxa small and triangular in shape, with three rows of spinules on anterior surface and outer margin. Coxa bare, massive and rather trapezoidal in shape. Basis wider than long, with few spinules at base of outer plumose spiniform seta. Endopod and exopod three-segmented, exopod longer than endopod. All segments of endopod with row of spinules along outer margin; enp-1 with one inner plumose seta; enp-2 with two inner plumose setae of which proximal short; enp-3 with two inner and two apical plumose setae, and one outer bipinnate spine. All segments of exopod with row of spinules along outer margin; exp-1 and exp-2 with one outer strong bipinnate outer spine, and one inner plumose seta; exp-3 with two inner and two distal plumose setae, and three outer bipinnate spines.

P3 (Figure 5A). As P2, except (1) intercoxal sclerite without spinules on two lateral blunt projections; (2) coxa with two rows of spinules; (3) basis with one outer bare seta; (4) enp-2 with only one inner plumose seta; (5) enp-3 with three inner plumose setae; (6) exp-3 bearing three inner plumose setae.

P4 (Figure 5B). Similar to P3, except (1) all segments more slender; (2) coxa with one row of spinules on outer margin; (3) enp-3 with only two inner plumose setae.

Armature formulae of female P1–P4 given in Table 2.

**Table 2.** Armature formulae of female P1–P4.

| Setal Formulae | Exopod | Endopod |
| :---: | :---: | :---: |
| P1 | 0.1.122 | 1.1.121 |
| P2 | 1.1.223 | 1.2.221 |
| P3 | 1.1.323 | 1.1.321 |
| P4 | 1.1.323 | 1.1.221 |

P5 (Figure 5C). Baseoendopod and exopod foliaceous, reaching the distal margin of genital-double somite. Baseoendopod with several rows of transversal fine spinules on surface, with outer slender seta on long cylindrical setophore, with some spinules and five setae at distal end, of which inner one and outer one naked, median three setae pinnate, with two tube pores proximally and one tube pore apically. Exopod oval, slightly

shorter than baseoendopod, ornamented with spinulose margin, with several rows of transversal fine spinules on surface, bearing six setae, of which second and third outermost slightly bulbous.

P6 (Figure 1D). Represented by a small segment bearing three smooth setae of unequal length.

### 3.2.2. Description of Male

Total body length ranging from 805 to 1071 μm (mean = 906 μm; $n$ = 3), greatest width ranging from 212 to 271 μm (mean = 232 μm; $n$ = 3).

Habitus (Figure 6A,B). Similar to in female, but slightly smaller; urosomites 2 and 3 not fused, with rows of spinules on ventral surface as figured. Caudal ramus with some spinules at distal margin, without bulbous setae.

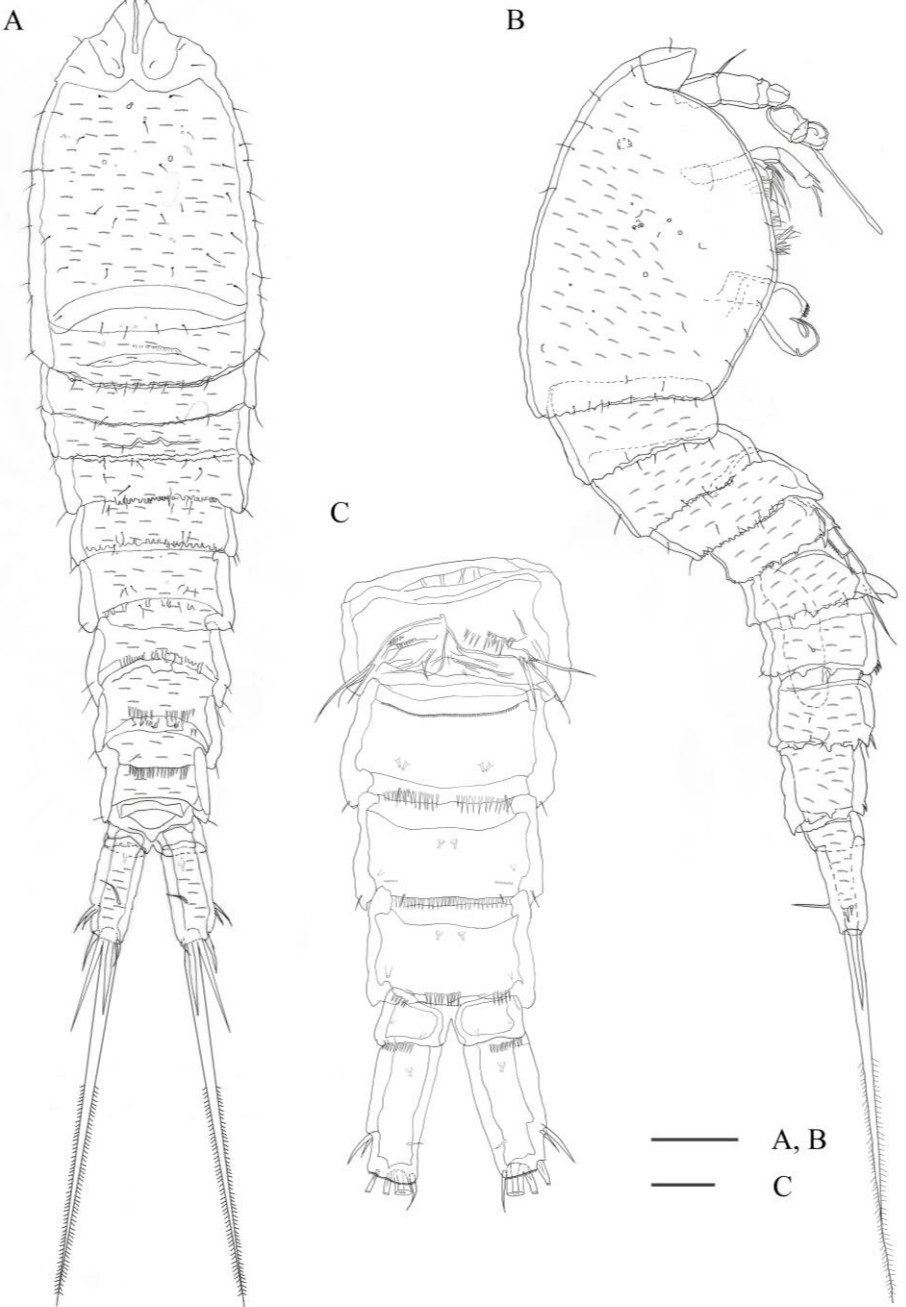

**Figure 6.** *Parathalestris xinzhengi* **sp. nov.**, male: (**A**) Habitus, dorsal. (**B**) Habitus, lateral. (**C**) Urosome, ventral. Scale bars: 10 μm (**A**,**B**); 50 μm (**C**).

Antennule (Figure 2C). Haplocer, ten-segmented, apical segment triangular in shape, with geniculation between seventh and eighth segments; first segment ornamented with spinules on anterior edge; with one aesthetasc on third, fourth and apical segments, respectively. Apex of apical segment recurved. Armature formula: 1-[1], 2-[11], 3-[6+ (1 + ae)], 4-[2], 5-[5 + (1 + ae)], 6-[2], 7-[3], 8-[0], 9-[2], 10-[7 + (2+ae)].

Antenna, mandible, maxillule, maxilla, maxilliped, P3 and P4 (not shown) as in female.

P1 (Figure 4B). Coxa with more spinulose rows. Basis with one curved stout spine on inner margin. Otherwise as in female.

P2 (Figure 4D). Endopod two-segmented. Enp-1 with one inner seta and row of spinules on outer margin; enp-2 strongly modified, with one short and two long plumose setae on inner margin, two unequal spines on outer margin, distal margin of segment produced into a sharp process, bearing one spine and one plumose seta. Exopod three-segmented; as in female.

P5 (Figure 5D). Baseoendopods fused medially, endopodal lobe short, extending to proximal third of exopod, with several rows of spinules on surface and three terminal elements, middle longest and innermost shortest. Exopod with six elements and with spinulose margin, outer apical one slender.

P6 (Figure 6C). Represented by two asymmetrical plates, one of these distinct, each leg bearing two long setae and one spine.

### 3.2.3. Molecular Results

The COI fragments (677 bp) were successfully amplified from two specimens of the new species. The COI sequence was submitted to GenBank finally (accession number: OQ831995).

### 3.2.4. Morphological Analysis Results

Three species most similar in morphology (*P. areolata*, *P. bulbiseta* and *P. xinzhengi* **sp. nov.**) cluster in the phylogenetic tree, and the morphological characteristic that supports their close relationship is the presence of bulbous seta on the caudal ramus (Character no. 2). They are sisters to *P. jacksoni* in the tree, which is supported by the concave inner edge of the maxilliped basis (Character no. 4). *P. croni*, *P. dovi* and the formers share the characteristic of short outermost seta on male P5 baseoendopod (Character no. 12). *P. cambriensis*, *P. harpactoides* and these six species form a clade supported by two shared characteristicss (Characteristics nos. 10 and 11). Additionally, the long outermost seta on the male P5 baseoendopod (Characteristic no. 12) seems to be an autapomorphy of *P. parviseta*, and *P. yeemini* can be separated from all the ingroup species by Characteristics nos. 7, 10 and 11 (Figure 7).

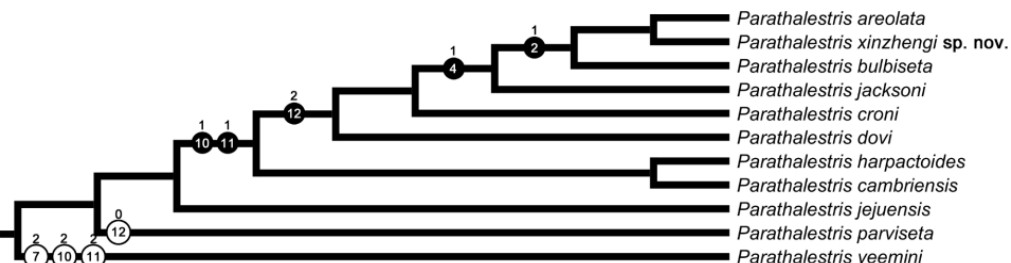

**Figure 7.** Phylogenetic tree inferred from 12 morphological characteristics (Table 1), constructed using MK+ASC models in IQ-Tree2 v.2.1.2. Full circles on the branch represent shared characteristics, empty circles represent unique characteristics. Number in the circle indicates characteristic number, and number above circle indicates characteristic state.

## 4. Discussion

The new species can be easily assigned to *Parathalestris* according to the following characteristics: (1) body fusiform and umbilicated from the side; (2) exp-2 and enp-1 of

P1 elongate, and exopod and endopod of P1 almost equal in length; (3) baseoendopod and exopod of female P5 foliaceous; (4) antennule of female nine-segmented, rostrum articulated with cephalothorax and expod of antenna two-segmented.

All valid species of *Parathalestris* are listed in Table 3, with comparisons of the most significant morphological characters. All morphological characteristics were collected from previous taxonomic publications. *Parathalestris xinzhengi* **sp. nov.** can be easily discriminated from other congeneric species by the combination of the following characteristics: caudal rami twice as long as wide, with caudal setae II and III bulbous; exopod of antenna two-segmented, with six setae; endites of maxilla long and slender; maxilliped concave; P1 exopod as long as endopod, enp-1 of P1 about 6.7 times as long as greatest width; female P5 exopod and baseoendopod with several rows of transversal fine spinules on surface, distal margin of exopod close to distal margin of baseoendopod, exopod bearing six elements; basis of male P1 with inner curved spine; exopod of male P5 with six elements.

Huys [7] subdivided the species of *Parathalestris* into three groups based on the length of P1 endopod and exopod. The new species *Parathalestris xinzhengi* **sp. nov.** belongs to species group I, of which the P1 exopod and endopod are of approximately the same length. In this group, *Parathalestris xinzhengi* **sp. nov.** is most closely related to *P. areolate* Itô Tat, 1972, sharing many similarities in the caudal ramus, setal arrangements of the antennary exopod, armature formula of P2–P4, shape of curved spine on male P1 basis and male P5. Significant differences between females of the two species include the ornamentation of the body (with many rows of microspinules in the new species, vs. areolated in *P. areolata*); relative length of P1 endopod (enp-1 of P1 about 6.7 times as long as greatest width in the new species, vs. 3.5 times in *P. areolata*) and relative length of P5 (distal margin of P5 exopod not exceeding the end of baseoendopod in the new species, vs. distal margin of P5 exopod overreaching the end of baseoendopod in *P. areolata*). The morphological differences between the new species and the other 11 species of this group are listed in Table 3.

Our phylogenetic analysis based on morphological data shows the new species locates at the most derived position in the tree of the species group I, together with *P. areolata* and *P. bulbiseta*, suggesting the bulbous seta on caudal ramus could be a specialized structure. However, this characteristic could be a homoplasy, as it is also shared by several species of other groups (e.g., *P. Hibernica*, Brady & Robertson, 1873, *P. intermedia* Gurney, 1930 and *P. infesta* Ho & Hong, 1988). Therefore, phylogenetic analysis integrating morphological and molecular data will be expected in the future to elucidate the relationships among the species of *Parathalestris* and the demarcation of different species groups.

So far, nine species of *Parathalestris* have been recorded from the West Pacific, including the new species. Our report of the new species is the first record of this genus from the China Seas. We speculate that the limited reports of *Parathalestris* from the China Seas are a consequence of insufficient sampling, and more intensive samplings will reveal the actual diversity of *Parathalestris* and other harpacticoid copepods in the future.

Lang [4] established a dichotomous key to species of *Parathalestris*, and Wells [3] gave a tabular key to 26 valid species. Huys [7] subdivided the species of *Parathalestris* into three groups but only gave a dichotomous key to the species group III. Below we present a key to the species group I, which is modified from the keys provided by Lang [4], Wells [3] and Huys [7].

**Table 3.** Morphological comparison among the species of *Parathalestris* Brady & Robertson, 1873.

| Group | I | | | | | | |
|---|---|---|---|---|---|---|---|
| Species | *P. croni* (Krøyer in Gaimard, 1842) | *P. harpactoides* (Claus, 1863) | *P. jacksoni* (Scott T., 1899) | *P. similis* Lang, 1936 | *P. irelandica* Roe, 1958 | *P. cambriensis* Wells, 1964 | *P. bulbiseta* Lang, 1965 |
| Length-width ratio of female caudal ramus | ≈4 | ≈1 | ≈2 | ≤1 | <1 | ≤1 | ≤1, with one bulbous seta distally |
| Armature formula of A2 exopod | 2(1:1,3) | 2(2:1,2) | 2(2:1,2) | unknown | 2(2:1,2) | 2(2:1,2) | 2(2:1,2) |
| Inner edge of maxilliped basis | straight | straight | concave | straight or convex | straight | straight | concave |
| Length-width ratio of P1 endopod | ≈4 | ≈5.5 | ≈4.5 | ≈4 | ≈5 | ≈5.5 | ≈4.5 |
| Length of female P5 baseoendopod | about 1/3 the length of exopod | not exceeding distal end of exopod | about half the length of exopod | not exceeding distal end of exopod | not exceeding distal end of exopod | slightly exceeding distal end of exopod | not exceeding distal end of exopod |
| Rows of transversal spinules on female P5 baseoendopod | absent | absent | absent | absent | absent | absent | absent |
| Length and armature formula of female P5 | extending about mid-length along genital double-somite, 5/6 | exceeding mid-length but not reaching end of genital double-somite, 5/6 | exceeding mid-length but not reaching end of genital double-somite, 5/6 | 5/6 | reaching about mid-length along genital double-somite, 5/6 | exceeding mid-length but not reaching end of genital double-somite, 5/6 | exceeding mid-length but not reaching end of genital double-somite, 5/6 |
| Inner spine of male P1 basis | curved inward | unknown | unknown | unknown | very short, curved inward | curved inward | curved outward |
| Armature formula of male P5 | 3/6 | 3/6 | unknown | 3/6 | 3/6 | 3/6 | 3/6 |
| Outermost seta of male P5 baseoendopod | unknown | half-length of median | unknown | half-length of median | half-length of median | half-length of median | 1/3 length of median |
| References | [20,21] | [22,23] | [24,25] | [21] | [26] | [27] | [4] |

**Table 3.** *Cont.*

| Group | I | | | | | |
|---|---|---|---|---|---|---|
| Species | *P. dovi* Marcus, 1966 | *P. areolata* Itô Tat, 1972 | *P. vinosa* Pallares, 1975 | *P. parviseta* Chang & Song, 1997 | *P. jejuensis* Song & Hwang, 2010 | *P. xinzhengi* **sp. nov.** |
| Length-width ratio of female caudal ramus | ≤1 | 1.5–2.0, with bulbous setae laterodistally and distally | ≤1 | 1 | 0.7 | ≈2, with bulbous setae laterodistally and distally |
| Armature formula of A2 exopod | 2(2:1,3) | 2(2:1,3) | unknown | 2(2:1,3) | 2(2:1,3) | 2(2:1,3) |
| Inner edge of maxilliped basis | straight or convex | concave | straight | straight | straight | concave |
| Length-width ratio of P1 endopod | ≈6 | ≈3.5 | ≈4 | 4.75 | ≈3.5 | ≈6.7 |
| Length of female P5 baseoendopod | about 2/3 length of exopod | about 2/3 length of exopod | unknown | not exceeding distal end of exopod | not exceeding distal end of exopod | slightly exceeding distal end of exopod |
| Rows of transversal spinules on female P5 baseoendopod | absent | present | unknown | present | absent | present |
| Length and armature formula of female P5 | reaching about mid-length along genital double-somite, 5/6 | exceeding mid-length but not reaching end of genital double-somite, 5/6 | unknown | reaching end of genital double-somite, 5/6 | reaching end of genital double-somite, 5/6 | reaching end of genital double-somite, 5/6 |
| Inner spine of male P1 basis | curved | curved | unknown | curved outward | normal | curved |
| Armature formula of male P5 | 3/6 | 3/6 | 3/6 | 3/5 | 3/5 | 3/6 |
| Outermost seta of male P5 baseoendopod | 1/3 length of median | 1/4 length of median | unknown | as long as median weak | half-length of median weak | 1/3 length of median |
| References | [3,28–30] | [1,10] | [3] | [1] | [2] | This paper |

**Table 3.** *Cont.*

| Group | II | | | | | | |
|---|---|---|---|---|---|---|---|
| Species | *P. clausii* (Norman, 1869) | *P. 16ibernica* (Brady & Robertson, 1873) | *P. intermedia* Gurney, 1930 | *P. incerta* Lang, 1936 | *P. para-harpactoides* Lang, 1939 | *P. aurantiaca* Pallares, 1975 | *P. patagonica* Pallares, 1975 |
| Length-width ratio of female caudal ramus | <1 | ≤1, with bulbous seta distally | ≤1, with bulbous seta laterodistally | unknown | <1 | unknown | unknown |
| Armature formula of A2 exopod | 2(2:1,2) | unknown | 2(2:1,3) | unknown | unknown | unknown | unknown |
| Inner edge of maxilliped basis | convex | concave | straight | concave | straight or convex | concave | concave |
| Length-width ratio of P1 endopod | ≈4–5 | ≈4–5 | >6 | ≈4–5 | ≈4–5 | ≈4–6 | ≈4–5 |
| Length of female P5 baseoendopod | not exceeding distal end of exopod | slightly exceeding distal end of exopod | not exceeding distal end of exopod | unknown | unknown | unknown | unknown |
| Rows of transversal spinules on female P5 baseoendopod | with only one row of transverse row spinules | absent | absent | unknown | absent | absent | absent |
| Length and armature formula of female P5 | exceeding mid-length but not reaching end of genital double-somite, 5/6 | exceeding mid-length but not reaching end of genital double-somite, 5/6 | 5/6 | unknown | 6/7 | 5/6 | 5/6 |
| Inner spine of male P1 basis | very short, curved inward | unknown | curved inward | very short, sharp and curved | unknown | unknown | very long, blunt and weakly curved |
| Armature formula of male P5 | 3/6 | 3/5 | 3/6 | 3/6 | 3/6 | 3/7 | 3/6 |
| Outermost seta of male P5 baseoendopod | as long as median | half-length of median | 2/3 length of median | slightly shorter than median | unknown | unknown | unknown |
| References | [22,24] | [21,24] | [31–33] | [3,21] | [3,21] | [3] | [3] |

**Table 3.** *Cont.*

| Group | III | | | | | | | | |
|---|---|---|---|---|---|---|---|---|---|
| Species | *P. affinis* Scott T., 1912 | *P. coatsi* Scott T., 1912 | *P. ganio* Brehm, 1938 | *P. californica* Lang, 1965 | *P. verrucosa* Itô Tat, 1970 | *P. pacificus* Chislenko, 1971 | *P. infesta* Ho & Hong, 1988 | *P. mourei* Masunari, 1988 | *P. yeemini* Chullasorn, Kangtia & Song, 2016 |
| Length-width ratio of female caudal ramus | ≤1 | ≤1 | 1.3 | ≤1 | 1.5–1.9 | ≈1 | 1, with bulbous seta laterodistally | 3 | <1 |
| Armature formula of A2 exopod | 2(2:1,1) | 2(0:3,3) | unknown | 2(2:1,3) | 2(2:1,4)/2,4 | 2(2:1,3) | 1–2(1:1,3) | 2(2:1,3) | 2(2:1,3) |
| Inner edge of maxilliped basis | straight | convex | unknown | concave | concave | concave | concave | straight | straight |
| Length-width ratio of P1 endopod | ≈3 | ≈3 | ≈4 | ≈5 | ≈4 | ≈6 | ≈5 | ≈3.5 | ≈3.34 |
| Length of female P5 baseoendopod | about 2/3 length of exopod | slightly exceeding mid-length of exopod | unknown | about 4/5 length of exopod | not exceeding distal end of exopod | slightly exceeding distal end of exopod | exceeding distal end of exopod | exceeding distal end of exopod | slightly exceeding distal end of exopod |
| Rows of transversal spinules on female P5 baseoendopod | absent | absent | unknown | present | absent | absent | absent | absent | present |
| Length and armature formula of female P5 | unknown | unknown | unknown | reaching at least last urosomite, 5/6 | exceeding mid-length but not reaching end of genital double-somite, 5/6 | reaching at least last urosomite, 5/6 | reaching about mid-length along genital double-somite, 5/6 | reaching distal margin of genital double-somite, 5/6 | exceeding distal margin of genital double-somite, close to penultimate somite, 5/6 |
| Inner spine of male P1 basis | unknown | unknown | unknown | curved inward | normal | unknown | transformed into a heavily sclerotized L-shaped bar | curved inward | normal |
| Armature formula of male P5 | unknown | unknown | 3/7 | 3/6 | 3/5 | 3/6 | 3/6 | 3/5 | 2/5 |
| Outermost seta of male P5 baseoendopod | unknown | unknown | unknown | overreaching half-length of median | half-length of median | 2/3 length of median | 3/4 length of median | as long as median weak | / |
| References | [34] | [34] | [3] | [3,21] | [1,9,15] | [31] | [6] | [35] | [8] |

Key to the species group I of *Parathalestris* Brady & Robertson, 1873:

1. Caudal ramus 4 times as long as wide ... ... ... ... ... ... *P. croni* (Krøyer in Gaimard, 1842)
   Caudal ramus less than 4 times as long as wide ... ... ... ... ... ... ... ... ... ... ... 2
2. Caudal ramus 2 times as long as wide ... ... ... ... ... ... ... ... ... ... ... ... 3
   Caudal ramus as long as width or shorter than wide ... ... ... ... ... ... ... ... 5
3. Caudal ramus with bulbous seta ... ... ... ... ... ... ... ... ... ... ... ... ... ... 4
   Caudal ramus without bulbous seta ... ... ... ... ... ... ... *P. jacksoni* (Scott T., 1899)
4. P1 enp-1 3.5 times as long as wide, female P5 exopod reaching beyond distal margin
   of baseoendopod ... ... ... ... ... ... ... ... ... ... ... ... *P. areolata* Itô, 1972
   P1 enp-1 6.7 times as long as wide, female P5 exopod reaching close to distal margin
   of baseoendopod ... ... ... ... ... ... ... ... ... ... ... ... **P. xinzhengi sp. nov.**
5. Inner edge of basis of maxilliped concave, with an obtuse angle close to proximal
   margin, caudal ramus with bulbous seta ... ... ... ... ... ... *P. bulbiseta* Lang, 1965
   Inner edge of basis of maxilliped straight or convex, caudal ramus without bulbous
   seta ... ... ... ... ... ... ... ... ... ... ... ... ... ... ... ... ... ... ... ... ... 6
6. P1 enp-1 more than 5 times as long as wide ... ... ... ... ... ... ... ... ... ... ... 7
   P1 enp-1 less than or about 5 times as long as wide ... ... ... ... ... ... ... ... 9
7. P1 enp-1 about 5 times as long as wide, female P5 beyond halfway but not beyond the
   end of the genital double-somite ... ... ... ... ... ... ... ... ... ... ... ... ... 8
   P1 enp-1 about 6 times as long as wide, female P5 extends only to about halfway
   along the genital double-somite ... ... ... ... ... ... ... ... ... *P. dovi* Marcus, 1966
8. Male P5 exopod rectangular, 2 times as long as wide; A2 exopod two-segmented, with
   setation formulae 2:1,2 ... ... ... ... ... ... ... ... ... ... ... *P. cambriensis* Wells, 1964
   Male P5 exopod pyriform, 1.5 times as long as wide; A2 exopod two-segmented, with
   setation formulae 1:1,3 ... ... ... ... ... ... ... ... ... *P. harpactoides* (Claus, 1863)
9. Exopod of male P5 baseoendopod with five setae ... ... ... ... ... ... ... ... ... 10
   Exopod of male P5 baseoendopod with six setae ... ... ... ... ... ... ... ... ... 11
10. Inner margin of male P1 basis with curved spine; the middle seta of the male P5
    baseoendopod minute ... ... ... ... ... ... ... ... ... *P. parviseta* Chang & Song, 1997
    Inner margin of male P1 basis without spine; the innermost seta of the male P5
    baseoendopod minute ... ... ... ... ... ... ... ... ... *P. jejuensis* Song & Hwang, 2010
11. P1 enp-1 about 4 times as long as wide ... ... ... ... ... ... ... ... ... ... ... ... 12
    P1 enp-1 about 5 times as long as wide ... ... ... ... ... ... ... *P. irelandica* Roe, 1958
12. Distal edge of male P2 Enp-2 with long, mucroniform projection and one stout,
    mucroniform spine on inner edge ... ... ... ... ... ... ... ... *P. vinosa* Pallares, 1975
    Distal edge of male P2 Enp-2 without project and one filiform seta on inner edge ...
    ... ... ... ... ... ... ... ... ... ... ... ... ... ... ... ... ... ... ... *P. similis* Lang, 1936

**Author Contributions:** Conceptualization, Q.K.; methodology, L.M. and Q.K.; molecular experiments and analysis, Y.W. and Q.K.; resources, all authors; data curation, all authors; writing—original draft preparation, Y.W. and L.M.; writing—review and editing, all authors. All authors have read and agreed to the published version of the manuscript.

**Funding:** This study was supported by the National Natural Science Foundation of China (No. 42276098) and Science & Technology Basic Resources Investigation Program of China (2017FY201404).

**Institutional Review Board Statement:** Not applicable.

**Data Availability Statement:** The data presented in this study are openly available in NCBI GenBank at https://www.ncbi.nlm.nih.gov/genbank/ accessed on 17 April 2023.

**Acknowledgments:** The authors thank Huilian Liu for her help gathering the specimens. The laboratory work was supported from Oceanographic Data Center, IOCAS.

**Conflicts of Interest:** The authors declare no conflict of interest.

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
