# Peer review of "Description of a New Species of Parathalestris Brady & Robertson, 1873 (Copepoda: Harpacticoida: Thalestridae) from China, with a Key to Its Affiliated Species Group†"

_diversity, doi:10.3390/d15040577_

Round 1

Reviewer 1 Report

This is a nice contribution to the knowledge of the diversity of the harpacticoid family Thalestridae, in which the authors described a new species of the genus Parathalestris. The manuscript is well-written and structured and the authors added some value to their manuscript with the addition of a key to the species of one of the previously defined species groups of the genus. Also, the figures are illustrative and of good quality. The tables also contain valuable information. I detected some errors which were highlighted in the pdf document. Briefly, and amongst other comments I suggest checking the general content of the abstract,  the species list in WoRMS (32 valid species vs 28 valid species in the manuscript), the use of the term "acrothek", and the labels of the figures. Also, I suggest adding some ljnes on the sampling methodology at the beginning of the section "Material and Methods". In my opinion, this manuscript deserves to be published after some minor corrections are done.

Author Response

Dear Reviewer,

Thank you very much for your time involved in reviewing the manuscript and your clear and detailed feedback. In the remainder of this letter, we discuss each of your comments individually along with our corresponding responses, and hope that the explanation has fully addressed all of your concerns.

Comment 1: Check the general content of the abstract, the species list in WoRMS (32 valid species vs 28 valid species in the manuscript), the use of the term "acrothek", and the labels of the figures.

Response 1: Thank you for the detailed review. We have carefully and thoroughly proofread the abstract and the labels of the figures. As for the valid species of the genus Parathalestris, we follow up on previous literature. Wells (2007) recorded 26 species belonging to Parathalestris, 27 species according Song & Hwang (2010) and 28 according Chullasorn et al. (2016). To avoid ambiguity, we replace acrothek with (setae number + ae). We also enlarge the illustrations for better recognization.

Comment 2: Adding sampling methodology at the beginning of the section "Material and Methods"

Response 2: Thanks for your suggestion on improving the integrity of the method. We have added the sampling methodology at the beginning of the section two. The relevant contents are provided below as a screen dump for your quick reference.

Comment 3: Just curious...is there any autapomorphy for the genus? Any synapomoprhy uniting all the species of Parathalestris?

Response 3: Thank you for your question. The characters of the genus listed in the introduction are according to Lang(1965), which not specified the autapomorphy or synapomorphy.

Comment 4: This might led to some confusion. The third urosomite is actually the second half of the genital double-somite. I think the third and fourth urosomites mentioned by the author are the fourth and fifth urosomites. Please check.

Response 4: Thanks for your careful review of the manuscript. The new species have totally nine segments and urosome five-segmented: the second segment was genital double-somite, which is only the two ventral ends are not fused. Therefore, we think the genital double-somite is a separate segment.

Comment 5: Antennule of Male. With only one geniculation? Are the distal segments geniculated?

Response 5: Thanks for your careful review of the manuscript and your question. After careful observed again, we think that male antennule has only one geniculation between seventh and eighth segments, and the distal segments shape into triangles.

We would like to take this opportunity to appreciate your help to us and this great opportunity for us to improve the manuscript. We hope you will be satisfied with the revised version.

Sincerely,

The Authors

Reviewer 2 Report

Present paper described a new species of Parathalestris from the Zhonggang wharf in Qindaom China. Authors suggested valuable description on the species with nicely presented illustrations. It has enough new findings to be published in Diversity, after few errors and additional information corrected.

1) Zoobank ID should be included for the paper and new taxa.

2) It would be better to give numbers for each caudal setae in the Figure 1.

3) In the Figure 3, the legend (B), and (C) should be changed. currently (B) is clearly male P1 not female as in the Figure.

4) Line 207 "1) All~" should be corrected to "1) all~"

5) Table 2, the title of the third column should be "Length-width ratio of female ~"

6) Table 2, the title of the 8th column should be " Rows of transversal~.

7) Line 361, Please give complete information for the reference, ex) volume numbers, pages, journal names, etc.

I understand that currently only few DNA marker sequences are available, however it would be better to suggest more details on mt COI data in the genus. Authors at least can compare the differences among the existing mt COI sequences of congeners, or species in the Family Thalestridae.

Author Response

Dear Reviewer,

Thank you very much for your care and patience involved in reviewing the manuscript and your very encouraging comments on the illustrations. We also appreciate your clear and detailed feedback and hope that the following explanation has fully addressed all of your concerns. To facilitate this discussion, we first retype your comments in italic font and then present our responses to the comments.

Comment 1: Zoobank ID should be included for the paper and new taxa.

Response 1: Thank you for your suggestion and we have uploaded the new taxa on the Zoobank and filled the ID in the manuscript.

Comment 2: It would be better to give numbers for each caudal setae in the Figure 1.

Response 2: Thank you for your great suggestion on improving the accessibility of our manuscript. We have added a detailed illustrations of caudal ramus with numbers for each seta. The relevant contents are provided below as a screen dump for your quick reference.

Comment 3: In the Figure 3, the legend (B), and (C) should be changed. currently (B) is clearly male P1 not female as in the Figure.

Response 3: Thanks for your careful review of the manuscript, we have corrected the legend in the Figure 3.

Comment 4: Line 207 "1) All~" should be corrected to "1) all~"

Response 4: We thank the reviewer for pointing this out. We have revised “All” to “all”.

Comment 5: Table 2, the title of the third column should be "Length-width ratio of female ~"

Response 5: This observation is correct. We have changed.

Comment 6: Table 2, the title of the 8th column should be " Rows of transversal~.

Response 6: We have changed the lowercase font to uppercase

Comment 7: Line 361, Please give complete information for the reference, ex) volume numbers, pages, journal names, etc.

Response 7: Thank you for the detailed review. We agree and have updated the reference.

Comment 8: I understand that currently only few DNA marker sequences are available, however it would be better to suggest more details on mt COI data in the genus. Authors at least can compare the differences among the existing mt COI sequences of congeners, or species in the Family Thalestridae.

Response 8: Thanks for your suggestion. But we found there are only few sequences related to this genus (two species and not under the same species group) in GenBank, which will be of little help to build a phylogenetic tree at present. So we add some phylogenetic analysis for species group I in the manuscript based on morphological data.

We would like to thank the referee again for taking the time to review our manuscript. We hope you will find this revised version satisfactory.

Sincerely,

The Authors

Reviewer 3 Report

The authors present the description of Parathalestris xinzhengi sp. nov. found at Zhonggang wharf in Qingdao, China. Together with a morphological description they provide some remarks regarding the affinities of the new species with its congeners, and a diagnostic key of the so-called “species group 1” of Parastenhelia.

The manuscript is generally well-written and absolutely worth for being published in diversity. However, although being no English native speaker, I found several potential slips of the pen and sometimes an infelicitous wording. Thus, I would recommend the revision of the manuscript by an English native speaker.

Abstract, introduction, and the Material and methods chapters are adequate and comprehensive. The illustrations are excellent but sometimes too small to recognize or discriminate between elements. Moreover, I suggest adding a larger illustration of one furcal ramus showing it more detailed, and with the setae labelled with Roman numerals (I–VII). The textual description, however, is often too vague and imprecise and does sometimes not even accord with the figures. Here, the authors have to check carefully text and illustrations again and correct errors and clear up ambiguities. I provide detailed comments in an uploaded revision of the manuscript.

The discussion is somewhat disappointing overall. I am aware that the authors do not claim a comprehensive phylogenetic analysis of Parathalestris. Nevertheless, it would have been desirable if they had undertaken a somewhat more detailed elaboration instead of a purely typological comparison of the new species with some other congeners. After all, with their description of P. xinzhengi sp. nov. they present a scientific hypothesis(!!); against this background, one might have expected that the authors would have looked for synapomorphies beyond a purely typological comparison of characters, which after all only indicates a diagnostic neighbouring position of single species listed on the basis of morphological similarities, but not a systematic relationship uncovered by means of detected morphological synapomorphies. Such a comparison would also have been possible without having to undertake a comprehensive revision of Parathalestris. This criticism may encourage the authors to revise their discussion once again. Nevertheless, it should not be understood in such a way that I demand a revision; the paper can also be published with the present discussion. In that case, however, it would be regrettable that the authors present a merely sufficient elaboration, which lies far below the possibilities.

I recommend acceptance of the manuscript after minor revision.

Author Response

Dear Reviewer,

Thank you very much for your time involved in reviewing the manuscript and your very encouraging comments on the merits. We also appreciate your clear and detailed feedback and hope that the explanation has fully addressed all of your concerns. In the remainder of this letter, we discuss each of your comments individually along with our corresponding responses. To facilitate this discussion, we first retype your comments in italic font and then present our responses to the comments.

Comment 1: Revise of the manuscript by an English native speaker.

Response 1: Thank you for the detailed review. We have carefully and thoroughly proofread the manuscript to correct all the grammar and typos.

Comment 2: The illustrations are excellent but sometimes too small to recognize or discriminate between elements. Moreover, I suggest adding a larger illustration of one furcal ramus showing it more detailed, and with the setae labelled with Roman numerals (I–VII).

Response 2: Thanks for your great suggestion on improving the accessibility of our manuscript. We have added a detailed illustration of furcal ramus. The relevant contents are provided below as a screen dump for your quick reference.

Comment 3: Nevertheless, it would have been desirable if they had undertaken a somewhat more detailed elaboration instead of a purely typological comparison of the new species with some other congeners.

Response 3: Thanks for your suggestion and we have included in the manuscript some phylogenetic analysis for species group I in the manuscript based on 12 morphological characters, showing the phylogeny in the species group I.

Comment 4: If the basis inner margin of the maxilliped is convex.

Response 4: Thank you for the detailed review. We think that the inner margin of the basis convex at proximal third and slightly concave at almost half-length, especially compared to P. yeemini, which inner margin is straight (Chullasorn et al., 2016).

(The image of the new species Mxp)

(The oral of P. yeemini, according to Chullasorn et al., 2016)

We would like to take this opportunity to thank you for all your time involved and this great opportunity for us to improve the manuscript. We hope you will find this revised version satisfactory.

Sincerely,

The Authors
